# Amnion Rupture Sequence

**DOI:** 10.3390/reports7020024

**Published:** 2024-03-27

**Authors:** Nicolae Gică, Florina Mihaela Nedelea, Livia Mihaela Apostol, Anca Maria Panaitescu, Iulia Huluță, Ana Maria Vayna, Radu Botezatu, Nicoleta Gana

**Affiliations:** 1Gynecology Department, Faculty of Medicine, Carol Davila University of Medicine and Pharmacy, 020021 Bucharest, Romania; gica.nicolae@umfcd.ro (N.G.); anca.panaitescu@umfcd.ro (A.M.P.); iuliahuluta16@gmail.com (I.H.); anamariavayna@gmail.com (A.M.V.); radu.botezatu@umfcd.ro (R.B.); gana_nicoleta@yahoo.com (N.G.); 2Clinical Hospital of Obstetrics and Gynaecology Filantropia, 011171 Bucharest, Romania

**Keywords:** amnion rupture sequence, encephalocele, neural tube defect

## Abstract

The amnion rupture sequence is a rare condition occurring early in pregnancy, resulting in complex fetal anomalies by disrupting normal embryonic development. The prevalence of amnion rupture sequence is reported to be 1.16 in 10,000 live births. This article explores the uncommon case of early amnion rupture leading to fetal encephalocele, suspected in the first trimester. Despite the variable and intricate nature of anomalies associated with this condition, cranio-facial and abdominal defects are frequently observed. Genetic testing was conducted, with normal results supporting our theory of amnion rupture. The patient decided to terminate the pregnancy, and the anatomopathological results confirmed the findings. This article discusses the diagnostic challenges, emphasizing the importance of timely identification through advanced imaging techniques.

## 1. Introduction

The amnion rupture sequence is an uncommon condition that occurs very early in pregnancy, and it leads to variable and complex fetal anomalies. Amnion parts constrict embryonic mass and interrupt the normal development of the anatomical structures [1]. The prevalence of this condition is 1.16 in 10,000 live births [2]. The anatomical parts affected depend on the time of amnion rupture [3]. Most of the cases described involve cranio-facial defects and abdominal defects [4]. There is no increased risk of recurrence in subsequent pregnancies [5].

## 2. Detailed Case Description

We present a case of a 35 year old woman who presented for her first-trimester ultrasound examination at 12 weeks and 1 day of gestation. This was her first pregnancy following spontaneous conception, and she had no history of miscarriages or any other medical issues. She was taking folic acid. Her first trimester was complicated by severe vaginal bleeding. The amniotic membrane wrapped the fetus, and the retro-amniotic space was largely increased. These findings were suggestive of an open neural tube defect due to an early amnion rupture. The most important sign suggesting the cause of the anomaly was the presence of the amniotic membrane attached to the fetus and the amniotic cavity being reduced in size.

We performed a transvaginal ultrasound to better visualize the spine and the brain. On the TV scan, we could demonstrate the occipital bone defect and the herniation of the fetal brain tissue throughout it (see Figure 1b). There were no other fetal anomalies detected on a thorough ultrasound examination.

A first trimester screening for chromosomal abnormalities and preeclampsia was made using a combination of maternal age, maternal characteristics and history, ultrasound markers such as nuchal translucency, tricuspid regurgitation, ductus venosus and nasal bone, and biochemistry (Beta-HCG and PAPP-A). The results showed a low chance for the most common trisomies (trisomy 21, 13 and 18) and a high chance for developing preeclampsia and fetal growth restriction. An invasive genetic (Whole Exome Sequencing) testing was performed, and it was negative. She was counselled for the outcome of this defect regardless of the genetic testing results, and she opted for the termination of pregnancy. She was admitted to the hospital. She was started on Mifepristone 200 mg per os and Misoprostol 0.2 mg^x^4 vaginally 36 h after the administration of Mifepristone. She aborted the fetus and the placenta in a few hours after the administration of Misoprostol. The anatomo-pathological results confirmed an occipital encephalocele measuring 1 cm and also described abnormally implanted ears, an abnormal shape of the chin and dysmorphic face features, with no other thoracic or abdominal anomalies. The histology confirmed our suspicion of amnion rupture, with thin amniotic strands being visible around the fetal cranium.

## 3. Discussion

The malformative syndrome known as congenital band sequence, amniotic rupture sequence, congenital ring constriction, amniotic band disruption complex, or congenital transverse defect is identified by various synonyms. These terms reflect the clinical diversity of the condition or the uncertainties regarding its cause [6]. The term amniotic constriction band encompasses a spectrum of congenital anomalies known for the significant variability in how they manifest clinically, from minor ring constrictions to severe defects that are not compatible with life [7].

Craniofacial defects are thought to arise from a vascular disruption sequence, which may or may not involve cephaloamniotic adhesion. However, certain chronological and topographical evidence does not align with the patterns of embryonic vascular development, casting doubt on the vascular hypothesis as the sole explanation. Nonetheless, a disruption in blood flow to the affected area, along with mechanical factors, could play a key role in the pathogenesis, particularly in cases of morphological disruption [6].

Amniotic rupture sequence is a highly uncommon condition. A widely accepted explanation for the disease’s development suggests that it starts with a rupture of the amniotic sac in the first trimester. This event leads to amniotic fluid leakage, which compresses the fetus and leads to the formation of fibrous bands. These bands can then directly harm fetal parts, causing amputation, deformation, or constriction. Another hypothesis suggests that the condition may arise from a disruption in the vascular supply to the embryo, serving as a potential cause [8].

The encephalocele is a cranial bone defect, with its most frequent location being in the occipital region. It can be associated with chromosomal or genetic abnormalities such as trisomy 13 and 18, Meckel–Gruber syndrome, Walker–Warburg syndrome and amniotic band syndrome [9]. The prognosis depends on the extension of the lesion. However, the mortality rate is about 50% for posterior encephalocele, and more than 50% of survivors have a neurological handicap [10].

The literature describes new-borns with encephalocele due to amniotic band syndromes [11,12,13,14]. There were also reported cases of encephaloceles diagnosed in the first trimester due to amniotic bands [15]. A study performed by Ushakov on 28 cases described the ultrasound features of amniotic band syndrome in the first trimester. He concluded that amniotic bands disappear with the progression of the pregnancy and that in many cases the bands could not be evaluated [15].

The significance of this case lies in its rarity and the insight it provides into the etiology of encephaloceles. Traditionally, the literature has focused on genetic mutations, nutritional deficiencies (such as folic acid), and environmental factors as primary causes of neural tube defects. However, this case underscores the mechanical disruptions that amniotic bands can cause, leading to significant cranial and neural anomalies. This case contributes to the medical literature by emphasizing the need for awareness and understanding of less common causes of neural tube defects such as encephalocele. It suggests that in cases of cranial anomalies detected prenatally, clinicians should consider the possibility of amniotic band syndrome as a contributing factor.

In conclusion, encephaloceles can be diagnosed in the first trimester of pregnancy during the 11–14 week scan [13,15]. The etiology should be investigated whenever possible in order to counsel the patient for current and future pregnancies.

## Figures and Tables

**Figure 1 reports-07-00024-f001:**
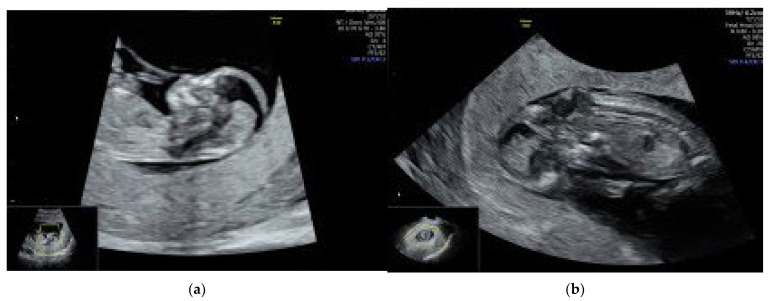
On ultrasound examination, we found a fetus with a CRL of 56.1 mm corresponding to 12 weeks and 1 day, a nuchal translucency of 1.4 mm and an abnormal posterior fossa of the fetal brain. In mid-sagital view of the fetus, we could not demonstrate a normal brainstem to brainstem-occipital bone ratio. The intracranial translucency was not visible. (**a**) Mid-sagital view of the fetal profile; the amniotic membrane was wrapping the fetus, and the retroamniotic space was largely increased. (**b**) Transvaginal (TV) ultrasound scan demonstrating the occipital bone defect and the herniation of the fetal brain tissue throughout it.

## Data Availability

Data are contained within the article.

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
