# Peer review of "Amnion Rupture Sequence"

_reports, 2024, doi:10.3390/reports7020024_

Round 1

Reviewer 1 Report

Comments and Suggestions for Authors

In this report, the authors reported a case related to amnion rupture sequence in pregnancy. No chromosomal or genetic abnormalities have been detected during the pregnancy. Below are a number of issues that the authors shall address or revise:

1. Authors have checked about chromosomal or genetic abnormalities of the fetus. I wonder whether these abnormalities of this woman could lead to this phenomenon.

2. Could authors present other details about this woman during pregnancy? Could the authors make some assumptions about the reasons for this case?

Comments on the Quality of English Language

1. Lines 12-14 were repeated in lines 16-17.

2. Lin2 28, 1.16 should be a dot.

3. Line 36. There should be a stop between “increased” and “These”.

Author Response

1. Thank you for your comments about our case. We have checked for chromosomal and genetic abnormalities  despite the low chance in this situation , because it was the first pregnancy of this women and she wanted to exclude a genetic condition that could affect her future pregnancies. It is known that encephalocele is a defect found in trisomy 13 and 18 or genetic syndromes such as Meckel-Gruber syndrome (autosomal recessive; polydactyly, multicystic kidneys, occipital cephalocele), Walker-Warburg syndrome (autosomal recessive; type II lissencephaly, agenesis of corpus callosum, cerebellar malformations, cataract).

2. In her first weeks of pregnancy she had severe vaginal bleeding. Many times the exact cause for amniotic rupture is not found. Sometimes patients have history of invasive testing or medication such as misoprostol. She was taking only folic acid.  We can speculate in this case that fetal vascular supply was restricted and led to amnion rupture.

We corrected the English mistakes and the punctuation as you suggested.

Reviewer 2 Report

Comments and Suggestions for Authors

I recommend to accept the article after minor revision.

Another comments

 Lines 59-61

 She was counselled for the outcome of this defect regardless of the genetic testing results and she opted for the termination of pregnancy. She was admitted to the hospital.

Note:

How was the pregnancy terminated? How long did the termination take? It should be presented in the case report description.

Lines 66-68

It can be associated with chromosomal or genetic abnormalities such as trisomy 13 and 18, Meckel-Gruber syndrome, Walker-Warburg syndrome and amniotic band syndrome.

Note:

Amniotic rupture sequence is a rare, non-recurrent disease complex and it is associated with defects of variable severity and at different sites, usually like a dismorphological pattern of congenital oro-craniofacial and limb defects. There are difficulties in the classification, which make diagnosis of unusual presentations rather difficult. Some authors call Amniotic rupture sequence like Amniotic band syndrome. This ambiguity in the nomenclature of the syndrome with different names should be more emphasized.

 Literature

ten Donkelaar HJ, Hamel BC, Hartman E, van Lier JA, Wesseling P. Intestinal mucosa on top of a rudimentary occipital meningocele in amniotic rupture sequence: disorganization-like syndrome, homeotic transformation, abnormal surface encounter or endoectodermal adhesion? Clin Dysmorphol. 2002 Jan;11(1):9-13. doi: 10.1097/00019605-200201000-00002. PMID: 11826876.

Alvarez-Manassero D, Manassero-Morales G. Cráneo en trébol y fisura facial bilateral [Cloverleaf skull and bilateral facial clefts]. Rev Chil Pediatr. 2015 Sep-Oct;86(5):357-60. Spanish. doi: 10.1016/j.rchipe.2015.07.017. Epub 2015 Oct 9. PMID: 26454757.

Atiyeh BS, Moucharafieh R. An unusual amniotic rupture sequence with thoracoabdominal restricting band, low-set posterior hairline, and trapezius contracture. J Craniofac Surg. 2010 Sep;21(5):1400-3. doi: 10.1097/SCS.0b013e3181ebcd6b. PMID: 20818258.

Basha MJ, Nagnur MI, Mohiuddin MS, Mohiuddin MJ, Salman S, C Sunder S. Streeter's Syndrome of Lower Limb Associated with CTEV. J Orthop Case Rep. 2022 Dec;12(12):50-53. doi: 10.13107/jocr.2022.v12.i12.3460. PMID: 37056605; PMCID: PMC10088393.

Jamsheer A, Materna-Kiryluk A, Badura-Stronka M, Wiśniewska K, Wieckowska B, Mejnartowicz J, Balcar-Boroń A, Borszewska-Kornacka M, Czerwionka-Szaflarska M, Gajewska E, Godula-Stuglik U, Krawczynski M, Limon J, Rusin J, Sawulicka-Oleszczuk H, Szwałkiewicz-Warowicka E, Swietliński J, Walczak M, Latos-Bieleńska A. Comparative study of clinical characteristics of amniotic rupture sequence with and without body wall defect: further evidence for separation. Birth Defects Res A Clin Mol Teratol. 2009 Mar;85(3):211-5. doi: 10.1002/bdra.20555. PMID: 19180633.

Author Response

Thank you for your suggestions. We added the information about the termination of pregnancy by medication.  

She was admitted to the hospital. She was started on Mifepristone 200 mg per os and Misoprostol 0.2 mgx4 vaginally after 36 hours after Mifepristone. She aborted the fetus and the placenta in a few hours after the administration of Misoprostol.

We have also took into consideration your suggested literature and we added extra information regarding the nomenclature of this pathology.   The malformative syndrome known as congenital band sequence, amniotic rupture sequence, congenital ring constriction, amniotic band disruption complex, or congenital transverse defect is identified by various synonyms. These terms reflect the clinical diversity of the condition or the uncertainties regarding its cause. [6] The term amniotic constriction band encompasses a spectrum of congenital anomalies known for their significant variability in how they manifest clinically, from minor ring constrictions to severe defects that are not compatible with life. [7] Craniofacial defects are thought to arise from a vascular disruption sequence, which may or may not involve cephaloamniotic adhesion. However, certain chronological and topographical evidence does not align with the patterns of embryonic vascular development, casting doubt on the vascular hypothesis as the sole explanation. Nonetheless, a disruption in blood flow to the affected area, along with mechanical factors, could play a key role in the pathogenesis, particularly in cases of morphological disruption. [6]  Amniotic rupture sequence is a highly uncommon condition. A widely accepted explanation for the disease's development suggests that it starts with a rupture of the amniotic sac in the first trimester. This event leads to amniotic fluid leakage, which compresses the fetus and leads to the formation of fibrous bands. These bands can then directly harm fetal parts, causing amputation, deformation, or constriction. Another hypothesis suggests that the condition may arise from a disruption in the vascular supply to the embryo, serving as a potential cause. [8]

Reviewer 3 Report

Comments and Suggestions for Authors

The Authors describe a case of encephalocele due to amnionic rupture sequence. The case is overall interesting, but it should be better presented. 

The case description is overall confusing, mixed with clinical information and general knowledge regarding the encephalocele. They should be kept separated with a general introduction, the case presentation and discussion/conclusion.

The Authors should expand the discussion, as they don't mention if other cases of encephalocele in the first trimester due to amnionic bands are described in literature. They should also mention at which gestational age, at ultrasound, this diagnosis can be suspected. 

Pathology is briefly mentioned, are there any pictures that can be add? how was brain histology? Were amniotic bands and amnion placental disepithelization demonstrated at histology? 

Please add all of this information. 

Comments on the Quality of English Language

English language must be thoroughly revised. 

Author Response

Thank you for your suggestions. We have revised our manuscript and we added a few extra information. We separated as you suggested : introduction, case presentation and discussion.

The literature describes newborns with encephalocele due to amniotic band syndromes. [11,12,] There were also reported cases of encephaloceles diagnosed in the first trimester due to amniotic bands. [13] A study performed by Ushakov on 28 cases, described the ultrasound features of amniotic band syndrome in  the  first  trimester. He concluded that amniotic bands disappear with progression of the pregnancy and in many cases the bands could not be evaluated [14]

In conclusion, encephaloceles can be diagnosed in the first trimester of pregnancy during the 11 – 14 week scan. [13,14,] 

Unfortunately we do not have any registered pictures of histology. The examination was written by the pathologist.

Reviewer 4 Report

Comments and Suggestions for Authors

1. You have not provided evidence that that encephalocele is related to the amnio rupture as pathology did not indicate any amnion attachment to the fetus that was associated with the NTD.

2. Ultrasound as well can see the amnion attachment if present.

3. There could be two separate developmental defects the NTD and abnormal amnion-chorion development attacment is required for the sequence.  

Comments on the Quality of English Language

Adequate.

Author Response

Thank you for your comments.

  1. We are sorry for the misunderstanding here. The pathology did indicate the amnion attachment, we briefly mentioned in the abstract that pathology confirmed, but we indeed did not mentioned in the description of this case.
  2. Ultrasound can see in most of the times the amniotic bands , but it is not always easily visible in the first trimester, as they are very thin. [Ushakov, F. and Lia, C. (2017), P10.04: Amniotic band syndrome: first trimester diagnosis and classification. Ultrasound Obstet Gynecol, 50: 186-186. https://doi.org/10.1002/uog.18098 ]
  3. Our diagnosis was completely made after the pathological examination. We only suspected an amnion rupture in early pregnancy because of the normal extensive genetic testing( WES) and the fetus being wrapped by the amniotic membrane and not changing the position the whole examination. We raised the suspicion and confirmed postabortum.

Round 2

Reviewer 1 Report

Comments and Suggestions for Authors

I am satisfied with the author’s responses to my issues raised in my initial review. I recommend that the revised paper be accepted.

Author Response

Thank you very much.

Reviewer 2 Report

Comments and Suggestions for Authors

Dear authors,

I have no more other comments. I recommend to accept the article for publication

Author Response

Thank you very much

Reviewer 3 Report

Comments and Suggestions for Authors

The Authors did not particularly expand what previously requested. Are there any pathological or histological pictures to add? What does this case contribute to the current literature?  

Comments on the Quality of English Language

English language must be reviewed. 

Author Response

Dear reviewer

Thank you for your Comments and Suggestions

The Authors did not particularly expand what previously requested. Are there any pathological or histological pictures to add? What does this case contribute to the current literature? 

Unfortunately we do not have any registered pictures of histology. The examination was written by the anathomopathologist.

    The significance of this case lies in its rarity and the insight it provides into the etiology of encephaloceles. Traditionally, the literature has focused on genetic mutations, nutritional deficiencies (such as folic acid), and environmental factors as primary causes of neural tube defects. However, this case underscores the mechanical disruptions that amniotic bands can cause, leading to significant cranial and neural anomalies. This case contributes to the medical literature by emphasizing the need for awareness and understanding of less common causes of neural tube defects like encephalocele. It suggests that in cases of cranial anomalies detected prenatally, clinicians should consider the possibility of amniotic band syndrome as a contributing factor.

The literature describes newborns with encephalocele due to amniotic band syndromes. We cited a few articles:

da Silva AJF, Silva CSME, Mariano SCR. Amniotic band syndrome with double encephalocele: A case report. Surg Neurol Int. 2020 Dec 22;11:448. doi: 10.25259/SNI_454_2020. PMID: 33408933; PMCID: PMC7771411.

Sepulveda, W., Wong, A.E., Andreeva, E., Odegova, N., Martinez-Ten, P. and Meagher, S. (2015), Sonographic spectrum of first-trimester fetal cephalocele: review of 35 cases. Ultrasound Obstet Gynecol, 46: 29-33. https://doi.org/10.1002/uog.14661

Ushakov, F. and Lia, C. (2017), P10.04: Amniotic band syndrome: first trimester diagnosis and classification. Ultrasound Obstet Gynecol, 50: 186-186. https://doi.org/10.1002/uog.18098

Yazawa O, Hirokawa D, Okamoto K, Tanaka M, Shibasaki J, Sato H. A new phenotype of amniotic band syndrome with occipital encephalocele-like morphology: a case report. Childs Nerv Syst. 2022 Jul;38(7):1405-1408. doi: 10.1007/s00381-021-05406-2. Epub 2021 Nov 5. PMID: 34739550.